# Gender Differences in the Risk of Ischemic Heart Disease According to Healthcare Utilization and Medication Adherence among Newly Treated Korean Hypertensive Patients

**DOI:** 10.3390/ijerph18031274

**Published:** 2021-01-31

**Authors:** Jiae Shin, Dongwoo Ham, Hee Young Paik, Sangah Shin, Hyojee Joung

**Affiliations:** 1Department of Public Health Sciences, Graduate School of Public Health, Seoul National University, 1 Gwanak-ro, Gwanak-gu, Seoul 08826, Korea; quellein@snu.ac.kr; 2Institute of Health and Environment, Seoul National University, 1 Gwanak-ro, Gwanak-gu, Seoul 08826, Korea; dwhampch@snu.ac.kr; 3Center for Gendered Innovations in Science and Technology Research (GISTeR), Korea Federation of Women’s Science & Technology Associations, 22 Teheran-ro 7-gil, Gangnam-gu, Seoul 06130, Korea; hypaik@kofwst.org; 4Department of Food and Nutrition, School of Food Science and Technology, Chung-Ang University, 4726 Seodong-daero, Daedeok-myeon, Gyeonggi-do, Anseong 17546, Korea

**Keywords:** gender differences, ischemic heart disease, healthcare utilization, medication adherence, National Health Insurance Service

## Abstract

We aimed to investigate gender differences in ischemic heart disease (IHD) according to healthcare utilization and medication adherence among newly treated Korean hypertensive adults. The National Sample Cohort version 2.0 of the National Health Insurance Service was used for analysis. Newly treated hypertensive patients ≥ 20 years and without IHD in 2002 were selected from a population that underwent health examination during 2003–2006. Of those patients, 11,942 men and 11,193 women were analyzed and followed up for 10 years. We determined the association between IHD and healthcare utilization and medication adherence using the Cox proportional hazards model. Hypertensive women patients had a lower risk of IHD than men patients (hazard ratio [HR] = 0.93, 95% confidence interval [CI] 0.88–1.00). The IHD risk was increased in patients who visited healthcare providers > 12 times/person-year (HR = 2.97, 95% CI 2.79–3.17), paid high out-of-pocket expense/person-year (HR = 1.55, 95% CI 1.41–1.69), and had medication nonadherence (HR = 1.67, 95% CI 1.58–1.77). However, the risk was decreased in patients who used both urban and rural areas (HR 0.75, 95% CI 0.67–0.84) and mixed types of providers (HR = 0.93, CI 0.88–0.99). The risk of IHD was significantly different between men and women only in the visiting frequency to healthcare providers (men, HR = 3.21, 95% CI 2.93–3.52; women, HR = 2.78, 95% CI 2.53–3.04, *p* for interaction = 0.0188). In summary, the risk of IHD was similar according to healthcare utilization and medication adherence between men and women, except visiting frequency to healthcare providers.

## 1. Introduction

Hypertension is a well-known risk factor for ischemic heart disease (IHD) [1,2]. Moreover, the prevalence of hypertension grew steadily among Koreans aged ≥ 30 years during 2007–2018 (men: from 26.8% to 33.2%; women: from 21.7% to 23.1%) [3].

Healthcare utilization is critical for hypertension management, because patients require continuous treatment to control their high blood pressure (BP). Uncontrolled BP leads to the development of complications, including IHD and death [4,5,6,7]. According to the World Health Organization (WHO), controlled BP and adequate cholesterol levels can reduce the risk of IHD by at least 16% [8]. Furthermore, a meta-analysis also reported that the risk of coronary heart disease decreased by 17% with a 10 mmHg reduction in systolic BP (SBP) [9].

For both men and women, the frequency of visits to healthcare providers (at least 3 month follow-up) and types of antihypertensive medication are usually set based on the current guidelines for hypertension treatment [10,11,12]. However, the application of hypertension treatment, such as healthcare utilization (location or types of healthcare providers, number of healthcare visits, and medical expenditure) and medication adherence, could vary by gender. A US study on medical expenditure reported that men had a greater share of lifetime medical expenditure attributed to hypertension (men, $88,033; women, $40,960) [13]. Similarly, studies have also revealed that men had higher medication adherence than women (men, 70.5%; women, 68.8%) [14,15]. Conversely, among Korean hypertensive patients, women had a higher treatment rate than men (men, 64.3%; women, 60.1%), even though women undergoing treatment did not show a higher control rate than their male counterparts (men, 71.0%; women, 70.6%) [16].

Several studies have estimated the effect of healthcare utilization or medication adherence on the risk of complications due to hypertension in Koreans [4,17,18,19]. However, there is a lack of studies to examine the risk of IHD according to healthcare utilization and medication adherence concerning gender. We hypothesized that healthcare utilization and medication adherence are different between men and women, and these gender differences can affect the risk of IHD differently between men and women. Hence, we aimed to investigate gender differences in the risk of developing IHD according to healthcare utilization and medication adherence among newly treated hypertensive patients in the Korean population.

## 2. Materials and Methods

### 2.1. Data Source

The National Health Insurance Service-National Sample Cohort version 2.0 (NHIS-NSC 2.0) was used for analysis. The National Health Insurance Service (NHIS) has gathered health insurance records from healthcare providers since its establishment and constructed the NHIS-NSC 2.0 with a million individuals based on total Korean subscribers to the service in 2006. The participants of the NHIS-NSC 2.0 were chosen to represent the Korean population using 1476 constructed strata including age group, sex, and income level [20]. The NHIS-NSC 2.0 has five databases: birth and death, insurance eligibility, treatment (records of healthcare service, expense, and medication), general health examination, and clinic (information on healthcare providers) for the period 2002–2015. Need for participants’ consent was waived because the NHIS provided anonymized personal information of the cohort population. Additional details are described elsewhere [20,21,22].

This study was reviewed and approved by the Institutional Review Board of Seoul National University (IRB No. E1905/002-002) and was granted permission for using NHIS data for research purposes (NHIS-2017-2-587).

### 2.2. Study Patients

With the de-identified numbers of cohort participants, we merged the aforementioned five databases of the NHIS-NSC 2.0. Among the one-million population of the NHIS-NSC 2.0, 51,043 hypertensive patients were included in the study if they met the following criteria: (1) being ≥ 20 years old and having undergone a general health examination at least once during 2003–2006; (2) not having IHD (International Classification of Diseases, Tenth Revision [ICD-10] codes: I20–I25) [23] in 2002; (3) not presenting outliers (which are defined as values over a mean of ±3 standard deviations) [24] of body mass index (BMI), fasting blood glucose (FBG), SBP, diastolic BP (DBP), or total cholesterol; (4) not missing information on residential area, income level, insurance type, family history, BMI, FBG, SBP, DBP, total cholesterol, family history, dietary habits, smoking status, alcohol consumption, or physical activity; (5) being diagnosed with hypertension (ICD-10, codes: I10–I13 or I15) [25]; and (6) being under antihypertensive medication as defined by the WHO Anatomical Therapeutic Chemical codes C02, C03, and C07–C09 [26].

Patients were excluded if they met the following criteria: (1) being on antihypertensive medication in 2002 or having begun intake of antihypertensive medication after 2006; (2) having visited any health-care providers only one time during the study period, as medication possession ratio (MPR) cannot be calculated if there is only one visit; (3) having been diagnosed with IHD within 1 year after starting antihypertensive medication treatment—this was established to assess the effect of medications at least after one year; and (4) having any missing values of healthcare providers, out-of-pocket expenses, or MPR.

As shown in Figure 1, a total of 23,135 hypertensive patients (11,942 men and 11,193 women) were included in the final analysis. Person-time for each subject was defined as the period between the first prescription of antihypertensive medications and the end of the follow-up due to one of four situations: first diagnosis of IHD, death, loss of insurance eligibility, or if 10 years had passed after the first prescription of antihypertensive medications. The data were presented as a value of years.

### 2.3. Definition of the Disease

Hypertensive patients were defined based on ICD-10 codes (I10–I13 or I15) in the medical treatment database (DB) and received the first prescription of antihypertensive medication during 2003–2006. The incidence of IHD was identified based on ICD-10 codes (I20–I25) [23] using the medical treatment DB; moreover, the date of patients’ first IHD diagnosis was recorded.

### 2.4. Healthcare Utilization Variables and Medication Adherence

Healthcare utilization, which refers to how medical services were used [27], was defined based on four variables: (1) location or (2) types of healthcare providers that patients visited for prescription of antihypertensive medication, (3) mean number of healthcare visits for medical prescription per person-year, and (4) mean out-of-pocket medical service expenses per person-year. The location of healthcare providers was divided into three groups: urban (cities—including seven metropolises, namely, Seoul, Busan, Daegu, Daejeon, Gwangju, Incheon, and Ulsan), rural (countryside), and mixed (urban and rural healthcare providers). Healthcare providers were classified into three types according to the standards of the Ministry of Health and Welfare [28]: (1) tertiary general hospital (certified by the government among general hospitals) or general hospital (≥100 sickbeds), (2) hospital (30–99 sickbeds), clinic (<30 sickbeds) or public health center, and (3) mixed (using a combination of the previous two types). Based on data from the medical treatment DB, frequency of healthcare visits for medication and amount of out-of-pocket expenses for healthcare providers were categorized into three levels. Frequency was divided into <4 times, 4–12 times, and >12 times per person-year. Out-of-pocket medical expenses were defined as a portion of the payment that patients paid to the healthcare providers among total medical insurance costs and categorized as high (upper 30%), medium (31–70%), and low (lower 30%). The cutoff point of visiting frequency was established according to the hypertension guidelines for treatment [11]; the recommended follow-up was to be carried out every 1–3 months, or 4–12 times per year, after beginning antihypertensive treatment.

Medication adherence is the extent to which a patient’s medication-taking behavior is consistent with their assigned prescription [29]. The MPR was used as a surrogate index for medication adherence [30], and medication adherence was defined as having an MPR ≥ 80% [31,32]. The MPR was calculated as the number of days’ supply of medication divided by the number of total days between the first visit and the last visit to a healthcare provider during the study period [31]. The MPR in this study was capped at 100% and classified into two categories: medication adherence (MPR ≥ 80%) and medication nonadherence (MPR < 80%) [31,32].

### 2.5. Covariates

Covariates that were controlled for in the statistical models included patients’ demographic, socioeconomic, biochemical, and lifestyle characteristics. Age was estimated from the year of birth at the year of health examination and classified into three groups: 20–39, 40–59, and ≥60 years. Residential areas were categorized into three groups: metropolitan (seven metropolises, namely, Seoul, Busan, Daegu, Daejeon, Gwangju, Incheon, and Ulsan), urban (cities except metropolises), and rural (countryside) areas. Similarly, income level was classified into three groups: high (upper 30%), medium (31–70%), and low (lower 30% or medical-aid beneficiaries) after an estimation based on the percentiles of insurance fee [21]. Insurance types were classified into three groups: employee insured, self-employed insured, and medical-aid beneficiaries.

BMI and biochemical indices, including FBG, SBP, DBP, and total cholesterol, were determined from the general health examination records, and classified into two levels: normal and obese (BMI ≥ 25.0 kg/m^2^), normal and diabetic (FBG ≥ 126 mg/dL), controlled and uncontrolled (SBP ≥ 140 mmHg, or DBP ≥ 90 mmHg), and normal and dyslipidemic (total cholesterol ≥ 240 mg/dL), respectively. The Charlson comorbidity index score was calculated for each subject based on diseases [33,34] diagnosed during 2002–2006 and divided into three groups (0, 1–2, and ≥3 scores).

Family medical history and lifestyle data (including dietary habits, smoking, alcohol consumption, and physical activity) were collected from self-reported questionnaires and subsequently categorized into two groups. Family medical history was divided as “none”, or “≥1”, referring to the number of family members with cardiovascular diseases, hypertension, or type 2 diabetes. Dietary habits were classified either as balanced and unbalanced [35]. The smoking category comprised two groups: “never” and “ever” (including former and current smokers) [36]. Alcohol consumption frequency was categorized as “none” or “>1/month”. Finally, physical activity frequency was divided into “none” and “≥1/week”.

### 2.6. Statistical Analyses

We performed Chi-square tests for categorical variables and Student’s *t*-test for continuous variables to determine significant differences between men and women’s general characteristics, healthcare utilization, and MPRs. The association of IHD incidence with healthcare utilization or medication adherence was estimated using multivariate hazard ratios (HRs) from Cox proportional hazards regression models by sex. Moreover, the assumption of proportional hazards was evaluated and satisfied by examining Schoenfeld residuals for patients [37]. We adjusted the model with age (20–39, 40–59, and ≥60 years) using the “STRATA” statement and with confounding variables: sex (for total patients only), residential area (metropolitan, urban, and rural area), income level (low, medium and high income level), family history (no and ≥1), insurance type (employee insured, self-employed insured, and medical-aid beneficiaries), Charlson comorbidity index (0, 1–2, and ≥3 scores), BMI (continuous), FBG (continuous), BP (continuous), total cholesterol (continuous), dietary habits (balanced and unbalanced), smoking (never and ever), alcohol consumption (none and ≥1/month), and physical activity (none and ≥1/week). *p* values for interaction were obtained by a likelihood ratio test to compare Cox proportional hazard models with and without cross-product terms for IHD incidence.

Population attributable risk (PAR) and its 95% confidence interval (CI) were calculated using the following formula: PAR = 100 × [*p*(HR-1) ÷ [*p*(HR-1) + 1]] (%), where *p* is the prevalence of exposure, expressed as a percentage [38,39]. All statistical analyses were performed using SAS Enterprise Guide version 7.1 software (SAS Institute, Cary, NC, USA). A *p* < 0.05 was considered to indicate statistical significance.

## 3. Results

In total, 23,135 hypertensive patients had 183,824 person-years of follow-up. A total of 6549 IHD cases were identified, and the mean patient follow-up period was 7.9 person-years. The baseline characteristics of the hypertensive patients by sex are shown in Table 1. The proportion of men and women differed by age: more than half of the men (52.5%) were in their 40s and 50s, while women were in their 60s and over (50.2%).

More women lived in rural areas (18.0%), compared with men (13.0%). A higher proportion of women had low income (28.9%) and were covered under medical-aid beneficiaries (1.1%), compared with men (21.6% and 0.7%, respectively). Over half of the patients (men, 62.0%; women, 52.8%) had uncontrolled BP at baseline. Men were more likely to have obesity (BMI ≥ 25 kg/m^2^) and diabetes (FBG ≥ 126 mg/dL), than women (men: 47.5% and 10.1%; women: 44.8% and 7.2%, respectively). Dyslipidemia (total cholesterol ≥ 240 mg/dL) and comorbidities were more common among women than among men, although smoking and alcohol consumption were less common among women. Men were more likely to have a balanced diet and be physically active, compared with women.

### 3.1. Healthcare Utilization and Medication Adherence of Hypertensive Patients

Table 2 shows the four healthcare utilization characteristics and medication adherence using MPR by sex during the study period. Compared with women, men preferred visiting urban healthcare providers and were more likely to utilize tertiary/general hospitals. However, women were more likely to visit single type (63.9%) or mixed types of healthcare providers (30.1%) than men (62.8% and 28.8%, respectively). In addition, hypertensive women visited healthcare providers more frequently than hypertensive men per person-year (men, 7.5 times/person-year; women, 7.9 times/person-year, *p* < 0.0001). There was no significant difference in out-of-pocket expenses between men and women. Men had a higher proportion of medication adherence (defined as MPR ≥ 80%) than did women (men, 42.5%; women, 39.1%). The mean MPR of all hypertensive patients was 64.8%, and men had a higher MPR than women (65.5% and 64.1%, respectively, *p* = 0.0003).

### 3.2. Hazard Ratios of Ischemic Heart Disease

IHD risk among hypertensive patients, according to healthcare utilization and medication adherence, is presented in Table 3. Hypertensive patients who visited healthcare providers > 12 times/person-year (vs. 4–12 times/person-year) and tertiary/general hospitals (vs. hospital/clinic/health centers) were more likely to develop IHD. In general, IHD risk was also higher for the patients with high out-of-pocket expenses/person-year (vs. a low level of expenses/person-year) and medication nonadherence (vs. medication adherence), regardless of gender. However, patients visiting healthcare providers in both urban and rural areas and mixed types of healthcare providers were significantly less likely to present IHD (HR = 0.75, 95% CI 0.67–0.84; HR = 0.93, 95% CI 0.88–0.99, respectively). Thus, men and women showed similar trends in the risk of developing IHD according to healthcare utilization and medication adherence. However, women, but not men, using mixed types of healthcare providers were significantly less likely to develop IHD (HR = 0.91, 95% CI 0.84–0.99). In addition, men visiting healthcare providers > 12 times/person-year showed a greater risk of IHD than their women counterpart (HR = 3.21, 95% CI 2.93–3.52; HR = 2.78, 95% CI 2.53–3.04, *p* for interaction = 0.0188).

### 3.3. Population Attributable Risk of Increased Ischemic Heart Disease Risk

Table 4 shows the estimated PAR of increased IHD risk according to different variables. Nonadherence (MPR < 80%) was the most significant predictor of the disease among men and women (27.8% and 29.3%, respectively).

## 4. Discussion

Among newly treated hypertensive patients, women patients showed a lower risk of IHD than men patients in this study using a Korean national representative sample data, the NHIS-NSC 2.0. The risks of IHD were similar according to healthcare utilization and medication adherence between men and women, except visiting frequency to healthcare providers. The risk of IHD was increased in patients who visited healthcare providers > 12 times/person-year, paid high out-of-pocket expense/person-year, and had medication nonadherence (MPR < 80%). However, the risk of IHD was decreased in those who used both urban and rural areas and mixed types of providers. The significantly different risk of IHD between men and women hypertensive patients was shown only in the visiting frequency to the healthcare providers per person-year. The PAR of IHD was the highest with medication nonadherence, followed by visiting frequency and out-of-pocket expenses.

Both men and women visiting healthcare providers > 12 times/person-year had an increased risk of IHD compared with those who visited health care 4–12 times/year, as is recommended in the guidelines for hypertension treatment. The data of our study were adjusted for comorbidities; hence, healthcare visits for prescriptions were not entirely over-represented. We selected newly treated hypertensive patients with other comorbidities except for IHD. Those patients with the other diseases needed to visit healthcare providers frequently. Additionally, they could have checked their hypertension progress with medication at each visit to healthcare providers. The increased risk of IHD might be related to comorbidities among the patients visiting healthcare providers > 12 times/person-year.

Women usually use healthcare services more often [40]; however, a higher frequency of visits to healthcare providers does not necessarily mean that all visits are needed. As has been previously observed, Koreans choose tertiary hospitals with the expectation of better care than primary/secondary healthcare providers [41]. In the present study, exclusive tertiary/general hospital use had an adverse effect on the incidence of IHD, regardless of sex, compared with the use of other types of healthcare providers (hospital/clinic/health center).

The increased risk of IHD among the patients visiting healthcare providers > 12 times/person-year might be related to comorbidities and negative aspects of the Korean healthcare system. The patients with comorbidities were more likely to be referred to tertiary hospitals. Healthcare providers under a fee-for-service system might induce more visits to hospitals [42]. In addition, Korean patients would like to visit tertiary hospitals, even though they are required to pay more. Accordingly, the patients with more risk factors of IHD could visit tertiary or general hospitals more frequently with greater out-of-pocket expenses [43].

Patients in this study showed a decreased risk of developing IHD when visiting healthcare providers both in urban and rural areas, compared with visiting healthcare providers only in urban areas. Additionally, women who visited several types of healthcare providers were less likely to develop IHD, compared with those visiting only one type of healthcare provider. This result could be explained by factors associated with the patients’ preference for treatment; those patients with comorbidities reported to prefer visiting tertiary hospitals instead of primary or secondary healthcare providers. These patients were more satisfied with using tertiary care because tertiary hospitals are government-certified and provide more specialized medical care services, compared with other healthcare providers [44]. This preference and satisfaction might have led patients to visit various types or more than two locations of healthcare providers, because tertiary hospitals are usually located in metropolitan areas and require a referral from primary healthcare providers.

Some studies have confirmed that gender-dependent treatments are more beneficial to men compared with women [45]. However, in our study, any favorable aspect of health care utilization in men was not revealed regarding IHD incidence. Furthermore, the proportion of men who had medication adherence (42.5%) was similar to that reported in previous research (42.4%) [25]; however, a lower proportion of women had medication adherence (39.1%). Several previous studies have shown that women have a greater MPR and more uncontrolled BP than men [46,47,48,49]. However, the effect of nonadherence on IHD risk was similar between men and women in our study. We also confirmed the importance of medication adherence in protecting IHD incidence because medication nonadherence (MPR < 80%) was the most significant predictor of the risk of IHD in both men and women patients. In the management program for hypertensive patients, medication adherence needs to be strengthened and well managed to protect IHD incidence, as addressed in the previous studies [17,18,25].

Men visiting healthcare providers >12 times on average were more likely to exhibit IHD, compared with women. In other studies, hypertensive female patients had a higher proportion of obesity [7], diabetes [50], and worse BP control [50]. However, in our study, men had worse biochemical indices (uncontrolled BP, obesity, and diabetes) at baseline. This might have caused adverse effects on the incidence of IHD among men in our study. Consequently, we observed that women were less likely to develop IHD (HR = 0.93, 95% CI 0.88–0.995).

This study has some limitations. First, information on disease in the medical treatment DB might be incomplete or inaccurate; for greater reimbursement and easy access, certain chronic diseases might have been overestimated, which could have affected our selection of patients at baseline. To minimize overestimation of study patients, we selected our hypertensive patients who had a diagnosis and prescriptions of antihypertensive medication. Second, BP, FBG, and cholesterol were measured in various places as the data sources in this study were nationally based on the results of health examinations. This might have affected the results of the study analysis because these biochemical indices were used as covariates. According to the European Society of Cardiology working group on cellular biology of the heart, the effects of depression and anxiety, pregnancy, and menopause on the risk of IHD were greater in women than in men [51]. Education level was also an important factor for gender differences in hypertension treatment [52]. The NHIS-NSC 2.0 did not contain data of uninsured medical expenses, salt intake, history of pregnancy, and the start time of menopause. Accordingly, these factors associated with the risk of IHD could not be analyzed in the current study.

Despite these limitations, our study presents some advantages. To our knowledge, earlier studies have not researched gender-based differences in the effect of healthcare utilization and medication adherence regarding IHD incidence in Korea. With retrospective data, hypertensive patients were followed up for ten years according to their healthcare utilization. We analyzed the direct HR between sexes, stratified according to healthcare utilization variables and MPR. Our results can be scientific evidence of personalized medicine to prevent IHD among Korean adults, considering healthcare utilization and medication adherence along with sex.

## 5. Conclusions

Our study indicated that healthcare utilization and medication adherence were associated with the risk of IHD in newly treated hypertensive patients. The risk of IHD was similar according to healthcare utilization and medication adherence between men and women, except visiting frequency to healthcare providers. There is a need for further studies on factors influencing the risk of IHD between men and women, such as socioeconomic status, lifestyle changes, and psychological and biological factors among hypertensive patients.

## Figures and Tables

**Figure 1 ijerph-18-01274-f001:**
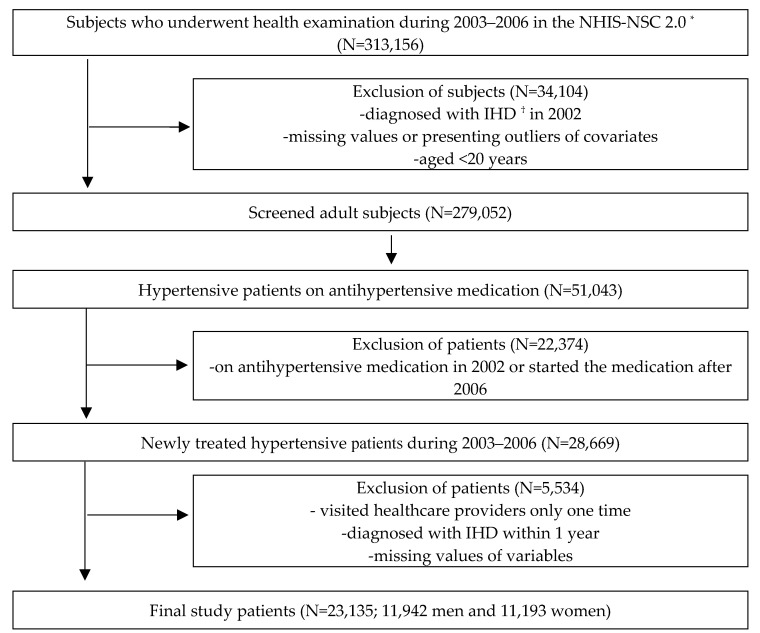
Flowchart of the selection process of study patients. ^*^ NHIS-NSC 2.0: National Health Insurance Service-National Sample Cohort version 2.0, ^†^ IHD: Ischemic heart disease.

**Table 1 ijerph-18-01274-t001:** General characteristics of hypertensive patients at baseline.

Variables	Total		Men		Women		*p*-Value ^1^
	N	%	N	%	N	%	
N	23,135		11,942	51.0	11,193	49.0	
Age (years)							
20–39	1498	6.5	1279	10.7	219	2.0	<0.0001
40–59	11,624	50.2	6268	52.5	5356	47.9	
≥60	10,013	43.3	4395	36.8	5618	50.2	
Residential area ^2^							
Metropolitan	9826	42.5	5246	43.9	4580	40.9	<0.0001
Urban	9743	42.1	5143	43.1	4600	41.1	
Rural	3566	15.4	1553	13.0	2013	18.0	
Income level ^3^							
Low	5815	25.1	2583	21.6	3232	28.9	<0.0001
Medium	8192	35.4	4391	36.8	3801	34.0	
High	9128	39.5	4968	41.6	4160	37.2	
Insurance type							
Employee insured	15,036	65.0	8244	69.0	6792	60.7	<0.0001
Self-employed insured	7891	34.1	3618	30.3	4273	38.2	
Medical-aid beneficiaries	208	0.9	80	0.7	128	1.1	
Family history ^4^							
No	18,798	81.3	9814	82.2	8984	80.3	0.0002
≥1	4337	18.7	2128	17.8	2209	19.7	
Hypertension status (mmHg)							
Controlled (<140/90)	9827	42.5	4541	38.0	5286	47.2	<0.0001
Uncontrolled (≥140/90)	13,308	57.5	7401	62.0	5907	52.8	
Systolic blood pressure (Mean ± SD)	137.33	16.0	138.53	15.6	136.06	16.4	<0.0001
Diastolic blood pressure (Mean ± SD)	85.03	10.7	86.63	10.8	83.33	10.3	<0.0001
Body mass index (kg/m^2^)							
<25	12,446	53.8	6268	52.5	6178	55.2	<0.0001
≥25	10,689	46.2	5674	47.5	5015	44.8	
(Mean ± SD)	24.77	3.0	24.84	3.0	24.68	3.0	<0.0001
Fasting blood glucose (mg/dL)							
<126	21,185	91.6	10,730	89.9	10,455	93.4	<0.0001
≥126	1950	8.4	1212	10.1	738	6.6	
(Mean ± SD)	97.83	19.6	99.72	20.8	95.82	17.9	<0.0001
Total cholesterol (mg/dL)							
<240	19,410	83.9	10,311	86.3	9099	81.3	<0.0001
≥240	3725	16.1	1631	13.7	2094	18.7	
(Mean ± SD)	203.10	37.3	199.38	36.7	207.07	37.6	<0.0001
Charlson comorbidity index score							
0	8714	37.7	5063	42.4	3651	32.6	<0.0001
1–2	6976	30.2	3516	29.4	3460	30.9	
3+	7445	32.2	3363	28.2	4082	36.5	
Dietary habits (Balanced)	17,014	73.5	9305	77.9	7709	68.9	<0.0001
Smoking (Ever)	6797	29.4	6385	53.5	412	3.7	<0.0001
Alcohol consumption (None)	13,669	59.1	4244	35.5	9425	84.2	<0.0001
Physical activity (None)	13,080	56.5	5797	48.5	7283	65.1	<0.0001

Note: Abbreviation: SD, standard deviation; ^1^
*p*-values were obtained from Chi-square test; ^2^ Metropolitan: metropolises (Seoul, Busan, Daegu, Daejeon, Gwangju, Incheon, and Ulsan), urban: cities, rural: countryside; ^3^ Low: lower 30 percentiles or medical-aid beneficiaries, medium: 31–70 percentiles, high: upper 30 percentiles; ^4^ ≥1: more than one of diabetes, hypertension, and heart disease in the recorded family history.

**Table 2 ijerph-18-01274-t002:** Healthcare utilization and medication adherence among hypertensive patients during the study period.

Variables		Total		Men		Women		*p*-Value ^1^
		N	%	N	%	N	%	
N		23,135		11,942	51.0	11,193	49.0	
Healthcare Utilization	Location of healthcare providers ^2^							
	Urban	19,306	83.4	10,215	85.5	9091	81.2	<0.0001
	Rural	1632	7.1	760	6.4	872	7.8	
	Mixed	2197	9.5	967	8.1	1230	11.0	
	Types of healthcare providers ^3^							
	Tertiary/general hospital	1683	7.3	1003	8.4	680	6.1	<0.0001
	Hospital/clinic/health center	14,654	63.3	7505	62.8	7149	63.9	
	Mixed	6798	29.4	3434	28.8	3364	30.1	
	Healthcare visits/person-year							
	<4 times	5980	25.8	3202	26.8	2778	24.8	<0.0001
	4–12 times	13,377	57.8	6929	58.0	6448	57.6	
	>12 times	3778	16.3	1811	15.2	1967	17.6	
	Mean ±SD	7.7	6.5	7.5	6.8	7.9	6.1	<0.0001
	Out-of-pocket expenses/person-year ^4^							
	Low	6940	30.0	3687	30.9	3253	29.1	0.0017
	Medium	9257	40.0	4658	39.0	4599	41.1	
	High	6938	30.0	3597	30.1	3341	29.8	
	Mean ± SD (1000 KRW)	24.1	24.2	24.0	25.2	24.1	23.1	0.6745
Medication adherence	Medication possession ratio							
	≥80%	9454	40.9	5075	42.5	4379	39.1	<0.0001
	<80%	13,681	59.1	6867	57.5	6814	60.9	
	Mean ± SD	64.8	28.8	65.5	29.0	64.1	28.6	0.0003

Abbreviation: KRW, Korean Won; SD, standard deviation; USD, United States dollars; KRW: the exchange rate on 5 June 2020 (1 USD ≈ 1203 KRW; ^1^
*p*-values were obtained from Chi-square test or *t*-test; ^2^ Urban: cities including metropolises (Seoul, Busan, Daegu, Daejeon, Gwangju, Incheon, and Ulsan), rural: countryside; ^3^ Tertiary general hospital: certified by the government among general hospitals, general hospital: ≥100 sickbeds, hospital: 30–99 sickbeds, clinic: <30 sickbeds, health center: no beds; ^4^ Low: lower 30 percentiles or medical-aid beneficiaries, medium: 31–70 percentiles, high: upper 30 percentiles.

**Table 3 ijerph-18-01274-t003:** Multivariate hazard ratios of the risk of ischemic heart disease according to healthcare utilization and medication adherence.

Variables	Total (n = 23,135)				Men (n = 11,533)				Women (n = 11,193)				*P* _interaction_ ^2^
	Cases	Person-Year	HR ^1^	95% CI	Cases	Person-Year	HR ^1^	95% CI	Cases	Person-Year	HR ^1^	95% CI	
Sex (Women vs. Men)	6549	183,824			3297	94,817	1.00	Ref	3252	89,007	0.93	(0.88, 1.00)	
Healthcare utilization & Medication adherence												
Location of Healthcare providers ^3^													
Urban	5446	153,404	1.00	Ref	2826	81,052	1.00	Ref	2620	72,352	1.00	Ref	
Rural	569	12,190	1.12	(0.98, 1.28)	255	5678	1.12	(0.92, 1.36)	314	6513	1.11	(0.93, 1.33)	
Mixed	534	18,230	0.75	(0.67, 0.84)	216	8088	0.77	(0.66, 0.91)	318	10,142	0.72	(0.62, 0.85)	0.2367
Types of healthcare providers ^4^													
Tertiary/general hospital	670	11,821	1.76	(1.61, 1.92)	395	7008	1.79	(1.59, 2.00)	275	4813	1.74	(1.52, 1.99)	
Hospital/clinic/health center	4024	116,425	1.00	Ref	1997	59,691	1.00	Ref	2027	56,734	1.00	Ref	
Mixed	1855	55,578	0.93	(0.88, 0.99)	905	28,118	0.96	(0.88, 1.04)	950	27,460	0.91	(0.84, 0.99)	0.4386
Healthcare visits/person-year													
<4 times	1513	48,613	1.03	(0.95, 1.11)	781	26,032	1.03	(0.92, 1.15)	732	22,581	1.02	(0.91, 1.15)	
4–12 times	3015	111,703	1.00	Ref	1523	57,784	1.00	Ref	1492	53,919	1.00	Ref	
>12 times	2021	23,508	2.97	(2.79, 3.17)	993	11,001	3.21	(2.93, 3.52)	1028	12,507	2.78	(2.53, 3.04)	0.0188
Out-of-pocket expenses/person-year ^5^												
Low	1710	56,558	1.00	Ref	881	30,005	1.00	Ref	829	26,553	1.00	Ref	
Medium	2156	76,823	0.94	(0.88, 1.02)	1051	38,674	0.95	(0.85, 1.05)	1105	38,148	0.94	(0.85, 1.05)	
High	2683	50,443	1.55	(1.41, 1.69)	1365	26,138	1.55	(1.37, 1.75)	1318	24,305	1.54	(1.36, 1.76)	0.1398
Medication possession ratio													
≥80%	2522	75,939	1.00	Ref	1347	40,484	1.00	Ref	1175	35,455	1.00	Ref	
<80%	4027	107,885	1.67	(1.58, 1.77)	1950	54,333	1.67	(1.55, 1.81)	2077	53,551	1.68	(1.55, 1.82)	0.2176

Abbreviations: HR, hazard ratio; CI, confidence interval; ^1^ Adjusted for age, sex (only for total patients), residential area, income level, insurance type, family history, Charlson comorbidity index, dietary habits, smoking, alcohol consumption, physical activity (categorical), body mass index, fasting blood glucose, systolic blood pressure, diastolic blood pressure, total cholesterol (continuous), and mutually adjusted for healthcare utilization variables and medication possession ratio (categorical); ^2^ P_interaction_: *p*-value of interaction between healthcare utilization and medication possession ratio and sex; ^3^ Urban: cities including metropolises (Seoul, Busan, Daegu, Daejeon, Gwangju, Incheon, and Ulsan), rural: countryside, mixed: both urban and rural areas; ^4^ Tertiary general hospital: certified by the government among general hospitals, general hospital: ≥100 sickbeds, hospital: 30–99 sickbeds, clinic: <30 sickbeds, and health center: no beds. ^5^ Low: lower 30 percentiles or medical-aid beneficiaries, medium: 31–70 percentiles, high: upper 30 percentiles.

**Table 4 ijerph-18-01274-t004:** Population attributable risks of increased ischemic heart disease risk according to healthcare utilization and medication adherence.

Variables	Total		Men		Women	
	PAR (%)	95% CI	PAR (%)	95% CI	PAR (%)	95% CI
Location of healthcare providers						
Rural only vs. Urban only	0.84	(−0.14, 1.94)	0.76	(−0.51, 2.24)	0.85	(−0.55, 2.51)
Types of healthcare providers ^1^						
Tertiary/general hospital only vs. hospital/clinic/health center only	5.24	(4.25, 6.27)	6.22	(4.72, 7.75)	4.30	(3.06, 5.67)
Healthcare visits/person-year						
>12 times vs. 4–12 times	24.34	(22.62, 26.16)	25.10	(22.64, 27.65)	23.83	(21.19, 26.39)
Out-of-pocket expenses/person-year ^2^						
High vs. Low	14.16	(10.95, 17.14)	14.21	(10.03, 18.43)	13.88	(9.70, 18.49)
Medication possession ratio						
<80% vs. ≥80%	28.38	(25.54, 31.29)	27.81	(24.03, 31.78)	29.28	(25.08, 33.30)

Abbreviations: PAR, population attributable risk; CI, confidence interval; ^1^ Tertiary general hospital: certified by the government among general hospitals, general hospital: ≥100 sickbeds; ^2^ Low: lower 30 percentiles or medical-aid beneficiaries, medium: 31–70 percentiles, high: upper 30 percentiles.

## Data Availability

Restrictions apply to the availability of these data. Data were obtained from the NHIS and are available from the National Health Insurance Sharing Service (at https://nhiss.nhis.or.kr/bd/ab/bdaba022eng.do) with the permission of the NHIS.

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
