# Peer review of "Gender Differences in the Risk of Ischemic Heart Disease According to Healthcare Utilization and Medication Adherence among Newly Treated Korean Hypertensive Patients"

_ijerph, 2021, doi:10.3390/ijerph18031274_

Round 1

Reviewer 1 Report

When analyzing the influence of gender as a variable differentiating the health condition of societies, it should be emphasized that this feature is taken into account in many research perspectives. Recognizing gender as one of the important determinants of inequality in health between women and men, attention should be paid to two types of dependence. The first one will correspond to the diversity resulting from biological reasons, the second one - from socio-cultural reasons. Moreover, gender differences in the pharmacokinetics of drugs are observed at the stages of absorption (absorption), distribution (disposition), metabolism (biotransformation) and elimination (removal) of a given drug. Thus, these differences can affect the course of treatment and how the body responds differently to taking the same medications. The Authors paid attention mainly to factors connected with care utilization and medication adherence between men and women. This is a very important and current problem therefore this manuscript can become an important source of information, However, before the publication the Authors should clarify some points:

  • The Authors should include conclusions in abstract.
  • Please explain the differences between men and women mentioned from line 47 to 56 of manuscript.
  • Factors included in table 1 are crucial for the course of cardiovascular diseases in women and men, if for each of these factors the groups of both sexes are not equally numerous, it may affect the reliability of the obtained results. It should be explained to what extent did the numerical inequality of the groups of both sexes affect the observed effects?
  • The discussion should be expanded. In its current form, it is not an explanation of the obtained results, but only a more detailed description of these results. Many of the issues contained in the discussion require clarification, among others: why both men and women visiting health care providers >12 times/person-year had an increased risk of IHD?
  • The Authors should explain why They claimed in conclusions that “there is a need for further studies on factors influencing the risk of IHD between sexes, such as socioeconomic levels, lifestyle change according to the guidelines for hypertension, or other psychological factors, among hypertensive patients”.

Author Response

Manuscript ID: ijerph-1089646

Title: Gender differences in the risk of ischemic heart disease according to health care utilization and medication adherence among newly treated Korean hypertensive patients

Reviewer #1:

When analyzing the influence of gender as a variable differentiating the health condition of societies, it should be emphasized that this feature is taken into account in many research perspectives. Recognizing gender as one of the important determinants of inequality in health between women and men, attention should be paid to two types of dependence. The first one will correspond to the diversity resulting from biological reasons, the second one - from socio-cultural reasons. Moreover, gender differences in the pharmacokinetics of drugs are observed at the stages of absorption (absorption), distribution (disposition), metabolism (biotransformation) and elimination (removal) of a given drug. Thus, these differences can affect the course of treatment and how the body responds differently to taking the same medications. The Authors paid attention mainly to factors connected with care utilization and medication adherence between men and women. This is a very important and current problem therefore this manuscript can become an important source of information, However, before the publication the Authors should clarify some points:

Response

-> We appreciate the opportunity to revise our manuscript. We carefully considered your opinions and revised the manuscript based on your comments. Detailed explanations regarding the changes we have made are described in the following responses.

  1. The Authors should include conclusions in abstract.

Response

-> Thank you for your thoughtful comments. As the reviewer commented, we have re-written the abstract including a more detailed conclusion. Furthermore, we have re-written the discussion part.

[Abstract] Lines 25–35: “Hypertensive women patients had a lower risk of IHD than men patients (hazard ratio [HR]=0.93, 95% confidence interval [CI] 0.88–1.00). The IHD risk was increased in patients who visited health care providers >12 times/person-year (HR=2.97, 95% CI 2.79–3.17), paid high out-of-pocket expense/person-year (HR=1.55, 95% CI 1.41–1.69), and had medication non-adherence (HR=1.67, 95% CI 1.58–1.77). However, the risk was decreased in patients who used both urban and rural areas (HR 0.75, 95% CI 0.67–0.84) and mixed types of providers (HR=0.93, CI 0.88–0.99). The risk of IHD was significantly different between men and women only in the visiting frequency to health care providers (men, HR=3.21, 95% CI 2.93–3.52; women, HR=2.78, 95% CI 2.53–3.04, p for interaction=0.0188). In summary, the risk of IHD was similar according to health care utilization and medication adherence between men and women, except visiting frequency to health care providers.

[Discussion] Lines 280–291: “Among newly treated hypertensive patients, women patients showed a lower risk of IHD than men patients in this study using a Korean national representative sample data, the NHIS-NSC 2.0. The risks of IHD were similar according to health care utilization and medication adherence between men and women, except visiting frequency to health care providers. The risk of IHD was increased in patients who visited health care providers >12 times/person-year, paid high out-of-pocket expense/person-year, and had medication non-adherence (MPR <80%). However, the risk of IHD was decreased in those who used both urban and rural areas and mixed types of providers. The significantly different risk of IHD between men and women hypertensive patients was shown only in the visiting frequency to the health care providers per person-year. The PARs of IHD was the highest with medication non-adherence, followed by visiting frequency and out-of-pocket expenses.

[Discussion] Lines 337–342: “We also confirmed the importance of medication adherence in protecting IHD incidence because medication non-adherence (MPR <80%) was the most significant predictor of the risk of IHD in both men and women patients. In the management program for hypertensive patients, medication adherence needs to be strengthened and well-managed to protect IHD incidence, as addressed in the previous studies [17,18,25].

  1. Please explain the differences between men and women mentioned from line 47 to 56 of manuscript.

Response

-> Thank you for your comments. In response, we explained more about the differences with substantial figures. We have re-written the sections.

Lines 55–61: “A US study on medical expenditure reported that men had a greater share of lifetime medical expenditure attributed to hypertension (men, $88,033; women, $40,960) [13]. Similarly, studies have also revealed that men had higher medication adherence than women (men, 70.5%; women, 68.8%) [14,15]. Conversely, among Korean hypertensive patients, women had a higher treatment rate than men (men, 64.3%; women, 60.1%), even though women undergoing treatment did not show a higher control rate than their male counterparts (men, 71.0%; women, 70.6%) [16].”

  1. Factors included in table 1 are crucial for the course of cardiovascular diseases in women and men, if for each of these factors the groups of both sexes are not equally numerous, it may affect the reliability of the obtained results. It should be explained to what extent did the numerical inequality of the groups of both sexes affect the observed effects?

Response

-> Thank you for your insightful comments. In our study, we performed our analysis stratified by sex. The total number of men and women was not significantly different (11,942 men and 11,193 women). We have also calculated the hazard ratio (HR) of IHD risk factors and the population attributable risk (PAR) to estimate the contribution of an individual factor on IHD by sex. We also adjusted our model with the known risk factors to reduce the effects of numerical inequality of our data on the result.

  1. The discussion should be expanded.In its current form, it is not an explanation of the obtained results, but only a more detailed description of these results. Many of the issues contained in the discussion require clarification, among others: why both men and women visiting health care providers >12 times/person-year had an increased risk of IHD?

Response

-> In accordance to the reviewer’s comments, we expanded our discussion to explain more about the frequency of visits to health care providers and the risk of IHD. We have re-written the discussion sections.

Lines 309–316: “The increased risk of IHD among the patients visiting health care providers >12 times/person-year might be related to comorbidities and negative aspects of Korean health care system. The patients with comorbidities were more likely to be referred to tertiary hospitals. Health care providers under fee-for-service system might induce more visits to hospitals [42]. In addition, Korean patients would like to visit tertiary hospitals, even though they are needed to pay more. Accordingly, the patients with more risk factors of IHD could visit tertiary or general hospitals more frequently with greater out-of-pocket expenses [43].”

  1. The Authors should explain why They claimed in conclusions that “there is a need for further studies on factors influencing the risk of IHD between sexes, such as socioeconomic levels, lifestyle change according to the guidelines for hypertension, or other psychological factors, among hypertensive patients”.

Response

-> We could not analyze such important gender-specific factors in our study as the data source (National Health Insurance Service National Sample Cohort version 2.0) did not include information of education level, pregnancy, and psychological factors. Therefore, we concluded the further studies are required with these factors to reflect gender differences in the risk of IHD such as depression and anxiety, pregnancy, and menopause. As you recommended, we have added more explanation in the limitation and re-written the conclusion section.    

Lines 358–364: “According to European Society of Cardiology working group on cellular biology of the heart, the effects of depression and anxiety, pregnancy, and menopause on the risk of IHD were greater in women than in men [51]. Education level was also an important factor for gender differences in hypertension treatment [52]. The NHIS-NSC 2.0 did not contain data of uninsured medical expenses, salt intake, history of pregnancy, and the start time of menopause. Accordingly, these factors associated with the risk of IHD could not be analyzed in the current study.”

Lines 375–380: “The risk of IHD was similar according to health care utilization and medication adherence between men and women, except visiting frequency to health care providers. There is a need for further studies on factors influencing the risk of IHD between men and women, such as socioeconomic status, lifestyle changes, and psychological and biological factors among hypertensive patients.”

We sincerely appreciate the reviewer for taking valuable time to provide useful comments and feedback. We are willing to respond to any further questions and/or comments that you may have.

Reviewer 2 Report

The current study aimed to investigate gender differences in ischemic heart disease (IHD) risk by health care utilization and medication adherence among newly treated Korean hypertensive patients. The study was based on a large sample from a national health survey in Korea and included 6549 cases from >23,000 patients followed up for an average of 7.5 years. The manuscript is well written, the methods and analyses were clearly described and presented, the manuscript contains lots of information, but the main downside is that it is not immediately clear what the key message from the study is. The authors could rethink what is unique about their study and elaborate on the public health implications of their main findings. They may want to revise the title and abstract to help guide the readers accordingly.

Main comments:

  1. Table 1 and 2 clearly show that demographic, clinical and lifestyle characteristics, health care utilization/ access and medication adherence differ between newly treated male and female hypertensive patients in Korea. However, gender does not appear to significantly alter the associations between health care utilization/ access, medication adherence, and IHD risk (Table 3,4). The evidence for gender as an effect modifier appears to be limited. The authors should make clear whether the main focus of the paper is about describing the health inequality (health care access and medication adherence) amongst male and female newly treated hypertensive patients, or is it to highlight the fact that health care utilization/ access and medication adherence affect IHD risk similarly in male and female hypertensive patients? The authors may want to revise the title and/or abstract to help guide readers accordingly.

  1. The other main issue is around interpretability of the study findings. Whilst I can imagine how the lack of medication adherence may increase IHD risk, it is not clear how increased access to health care (>12 times/ year), treatment in big hospital (tertiary hospitals), and higher out-of-pocket expenses would worsen health outcome. Could the authors discuss potential causal mechanism, and could the authors rule out residual confounding due to patient comorbidity?

  1. With the potential of confounding, do the authors think there may be merit in performing a nested case-control study in the same cohort population, matching for the key confounding variables?

  1. Line 181: The authors mention that age was stratified in the analyses, however, this is not presented in the results. Could the author please clarify.

  1. Lines 192: PAR=100 × [P(HR-1) ÷ [P(HR-1) + 1] (%). Please double check the equations quoted is corrected. I think one of the brackets is misplaced.

Author Response

The current study aimed to investigate gender differences in ischemic heart disease (IHD) risk by health care utilization and medication adherence among newly treated Korean hypertensive patients. The study was based on a large sample from a national health survey in Korea and included 6549 cases from >23,000 patients followed up for an average of 7.5 years. The manuscript is well written, the methods and analyses were clearly described and presented, the manuscript contains lots of information, but the main downside is that it is not immediately clear what the key message from the study is. The authors could rethink what is unique about their study and elaborate on the public health implications of their main findings. They may want to revise the title and abstract to help guide the readers accordingly.

Response

-> We appreciate the opportunity to revise our manuscript. We carefully considered your opinions and revised the manuscript based on your comments. Detailed explanations regarding the changes we have made are described in the following responses.

  1. Table 1 and 2 clearly show that demographic, clinical and lifestyle characteristics, health care utilization/ access and medication adherence differ between newly treated male and female hypertensive patients in Korea. However, gender does not appear to significantly alter the associations between health care utilization/ access, medication adherence, and IHD risk (Table 3,4). The evidence for gender as an effect modifier appears to be limited. The authors should make clear whether the main focus of the paper is about describing the health inequality (health care access and medication adherence) amongst male and female newly treated hypertensive patients, or is it to highlight the fact that health care utilization/ access and medication adherence affect IHD risk similarly in male and female hypertensive patients? The authors may want to revise the title and/or abstract to help guide readers accordingly.

Response

-> Thanks for your pertinent comment. Considering your comments, we added our hypothesis in the introduction to make our purpose clear.

Lines 65–67: “We hypothesized that health care utilization and medication adherence are different between men and women, and these gender differences can affect the risk of IHD differently between men and women.”

  1. The other main issue is around interpretability of the study findings. Whilst I can imagine how the lack of medication adherence may increase IHD risk, it is not clear how increased access to health care (>12 times/ year), treatment in big hospital (tertiary hospitals), and higher out-of-pocket expenses would worsen health outcome. Could the authors discuss potential causal mechanism, and could the authors rule out residual confounding due to patient comorbidity?

Response

-> Thank you for your valuable comment. Based on your comments, we expanded our discussion as below, considering potential causal mechanism. We adjusted our model with Charlson comorbidity index to rule out residual confounding due to patient comorbidity.  

Lines 309–316: “The increased risk of IHD among the patients visiting health care providers >12 times/person-year might be related to comorbidities and negative aspects of Korean health care system. The patients with comorbidities were more likely to be referred to tertiary hospitals. Health care providers under fee-for-service system might induce more visits to hospitals [42]. In addition, Korean patients would like to visit tertiary hospitals, even though they are needed to pay more. Accordingly, the patients with more risk factors of IHD could visit tertiary or general hospitals more frequently with greater out-of-pocket expenses [43].”

  1. 3. With the potential of confounding, do the authors think there may be merit in performing a nested case-control study in the same cohort population, matching for the key confounding variables?

Response

->Thank you for your valuable comment. A nested case-control study has advantages in minimizing selection bias and recall bias compared with a case-control study. However, matching for confounding factors is unlikely to estimate its effect in the analysis and there is a possibility of overmatching. Considering these limitations, we decided to take a patient-cohort design. Furthermore, the participants of our data source, the National Sample Cohort of the National Health Insurance Service (NHIS-NSC 2.0), are representative of the Korean population. We also added detailed information about our data source, the NHIS-NSC 2.0.

Lines 77–79: “The participants of the NHIS-NSC 2.0 were chosen to represent the Korean population using 1476 constructed strata including age group, sex, and income level [20].”

  1. Line 181: The authors mention that age was stratified in the analyses, however, this is not presented in the results. Could the author please clarify.

Response

->Thank you for your valuable comment. Age was ruled out by using the STRATA statement in the model. To make it clear, we have re-written the statistical analysis part.

Lines 189–190: “We adjusted the model with age (20–39, 40–59, and ≥60 years) using the "STRATA" statement and with confounding variables.

  1. Lines 192: PAR=100 × [P(HR-1) ÷ [P(HR-1) + 1] (%). Please double check the equations quoted is corrected. I think one of the brackets is misplaced.

Response

->Thank you for your thoughtful comment. Accordingly, we added one bracket at the end of the formula.

Line 201: “PAR=100 × [P(HR-1) ÷ [P(HR-1) + 1]] (%)”.

We sincerely appreciate the reviewer for taking valuable time to provide useful comments and feedback. We are willing to respond to any further questions and/or comments that you may have.

Reviewer 3 Report

The manuscript looks simple having on mind the research area but really it is very well written. The materials and methods section is described very simple and understandable for reader. The statistical analyses methods are correct and well described. Results of the study also are presented correctly and clear for readers of the manuscript. Conclusions are formulated according to the results of the study. I have no suggestions what change and correct in the manuscript and recommend it for publication

Author Response

The manuscript looks simple having on mind the research area but really it is very well written. The materials and methods section is described very simple and understandable for reader. The statistical analyses methods are correct and well described. Results of the study also are presented correctly and clear for readers of the manuscript. Conclusions are formulated according to the results of the study. I have no suggestions what change and correct in the manuscript and recommend it for publication

Response

-> We appreciate your comments on our manuscript. We carefully considered your opinions and checked again the manuscript based on your comments.

We sincerely appreciate the reviewer for taking your valuable time to provide useful comments and feedback. We are willing to respond to any further questions and/or comments that you may have.

Round 2

Reviewer 1 Report

No comments

Reviewer 2 Report

I have no further comments.